# Acute Kidney Injury in COVID-19 Patients: Pathogenesis, Clinical Characteristics, Therapy, and Mortality

**DOI:** 10.3390/diseases10030053

**Published:** 2022-08-19

**Authors:** Venice Chávez-Valencia, Citlalli Orizaga-de-la-Cruz, Francisco Alejandro Lagunas-Rangel

**Affiliations:** 1Department of Nephrology, Hospital General Regional Hospital No. 1, Instituto Mexicano del Seguro Social, Bosque de los Olivos No. 101. Av. La Goleta Mpo. Charo, Morelia 61301, Mexico; 2Department of Surgical Sciences, Uppsala University, 75124 Uppsala, Sweden

**Keywords:** AKI, SARS-CoV-2, ACE2, molecular mechanisms

## Abstract

Coronavirus disease 2019 (COVID-19) is a disease caused by infection with the SARS-CoV-2 virus and has represented one of the greatest challenges humanity has faced in recent years. The virus can infect a large number of organs, including the lungs and upper respiratory tract, brain, liver, kidneys, and intestines, among many others. Although the greatest damage occurs in the lungs, the kidneys are not exempt, and acute kidney injury (AKI) can occur in patients with COVID-19. Indeed, AKI is one of the most frequent and serious organic complications of COVID-19. The incidence of COVID-19 AKI varies widely, and the exact mechanisms of how the virus damages the kidney are still unknown. For this reason, the purpose of this review was to assess current findings on the pathogenesis, clinical features, therapy, and mortality of COVID-19 AKI.

## 1. Introduction

Coronavirus disease 2019 (COVID-19) is a pandemic disease that has posed a great challenge to the world. This disease is caused by infection with the SARS-CoV-2 virus and can be asymptomatic or cause a wide spectrum of symptoms including fever (70–90%), dry cough (60–86%), shortness of breath (53–80%), fatigue (38%), myalgia (15–44%), nausea/vomiting or diarrhea (15–39%), headache, weakness (25%), and runny nose (7%) [1]. Common complications among COVID-19 patients include pneumonia (75%), acute respiratory distress syndrome (15%), acute liver injury (19%), cardiac injury (7–17%), prothrombotic coagulopathy (10–25%), acute renal failure (9%), neurological manifestations (8%), acute cerebrovascular disease (3%), and shock (6%) [1,2].

Although the greatest damage occurs in the lungs, the kidneys are not exempt, and acute kidney injury (AKI) may occur in patients with COVID-19 [3]. Indeed, lung disease and AKI are among the most frequent and serious organic complications of COVID-19. According to the Risk, Injury, Failure, Loss, and End-stage renal disease (RIFLE) classification, acute kidney injury (AKI) was defined as an increase in creatinine ≥ 50% from baseline and/or a drop in glomerular filtration rate (GFR) of ≥25% and/or decreased urine output below 0.5 mL/kg/h for 6 h or more. The “acute” element of the AKI definition requires the ≥50% increase be known or assumed to have developed over ≤7 days [4]. Meanwhile, the Acute Kidney Injury Network (AKIN) group has proposed a modified version of the RIFLE criteria. In AKIN stage 1 (analogous to RIFLE-Risk), a minor 48-h change in serum creatinine of more than 0.3 mg/dL (≥26.2 μM/L) was suggested as the threshold for AKI. In addition, patients who received renal replacement therapy were reclassified as stage AKIN-3 (analogous to RIFLE-Failure) [5]. In the studies detailed below, both definitions were used [6,7,8].

The incidence of COVID-19 AKI varies between 0% and 56.9%, largely depending on whether the patients are in the intensive care unit (ICU) or not [9,10,11,12]. The literature reports that COVID-19 AKI is likely to affect >20% of hospitalized patients and >50% of ICU patients [13]. Notably, COVID-19 AKI carries adverse outcomes, including the development or worsening of comorbid conditions, as well as increased use of health care resources. In this way, the purpose of this article was to detail the main findings found in patients who developed COVID-19 AKI, with special emphasis on incidence, pathogenesis, pathology, associated clinical characteristics, therapy, and mortality.

## 2. SARS-CoV-2 Kidney Infection

The SARS-CoV-2 virus has four well-identified structural proteins: the spike glycoprotein (S), the small envelope glycoprotein (E), the membrane glycoprotein (M), and the nucleocapsid protein (N). The S protein is divided into two subunits called S1 and S2 and forms homotrimers that protrude from the viral surface [14]. Remarkably, the S protein mediates viral entry into the host cell by first binding to the angiotensin-converting enzyme 2 (ACE2) receptor through the receptor-binding domain (RBD) that is part of the S1 subunit, and then the subunit S2 fuses the membranes of the virus and the host cell. Meanwhile, the N protein binds to the RNA of the virus and participates in processes related to the viral replication cycle and the response of the infected cell. The M protein stabilizes the complex between the N protein and viral RNA and promotes the completion of viral assembly. Lastly, protein E plays a role in virus production and maturation [15].

It has been reported that the kidney is one of the organs that most expresses ACE2 and that the renal cells with the highest expression are the proximal tubular cells and, to a lesser extent, the podocytes [16,17,18]. SARS-CoV-2 protein S was also found to partially colocalize with ACE2 in tubules and parietal cells from kidney biopsy samples and in urine sediment cells [19]. Furthermore, postmortem analysis of COVID-19 patients found that they had ACE2 upregulation in the kidney relative to people without COVID-19 [18]. Differences in terms of sex should also be considered, since androgens promote the expression of TMPRSS2, which collaborates in the entry of the virus [20]. In this regard, it was reported in a rodent model of polycystic ovary syndrome (POCS) that androgens increased ACE2 protein expression in the kidney [21]. With all this background, it is clear that the kidney can be greatly infected and can also act as a reservoir for the virus (Figure 1) [16]. Indeed, reports from 21 autopsies mentioned the presence of viral RNA in 66.6% of the kidneys [22].

## 3. Pathophysiology of COVID-19 AKI

The pathophysiology of COVID-19 AKI is thought to be multifactorial and involves cardiovascular comorbidity, direct effects of the virus on the kidney, local and systemic inflammatory and immune responses, endothelial injury, and activation of coagulation pathways and the renin-angiotensin system (Figure 2) [23]. Pan X. et al. [24] mentioned that the cytopathic effects of SARS-CoV-2 on podocytes and proximal tubule cells might be associated with the development of COVID-19 AKI, especially in patients with sepsis. Likewise, the presence of viral particles in renal endothelial cells has been reported, suggesting that viremia may also cause endothelial damage that promotes vasoconstriction, a state of hypercoagulability, and macrophage activation leading to formation of microthrombi and renal microvasculature injury [18,24]. Furthermore, the loss of functional nephrons after injury could increase the development of renal fibrosis. Meanwhile, COVID-19 pneumonia can cause right ventricular failure and lead to renal congestion, while left ventricular dysfunction can lead to hypotension, decreased cardiac output, and hypoperfusion of the kidneys [25]. Severe COVID-19 can cause skeletal muscle damage leading to myoglobulin release, which induces kidney damage through pigment cast formation, causing tubular obstruction and tubular toxicity related to iron release [26].

In clinical studies, COVID-19 AKI has been associated first with sepsis (cytokine storm) and septic shock (hypoxia), second with the use of nephrotoxic drugs, and third with direct cell damage caused by SARS-CoV-2 [16,17]. Histopathological reports of AKI and COVID-19 are scarce and are usually autopsy series or isolated reports of some glomerulopathy. However, Su H. et al. [18] reported pathologic renal abnormalities in postmortem cases, demonstrating the presence of coronavirus clusters in the proximal tubular epithelium, podocytes, and distal tubules along with loss of brush border and non-isometric vacuolization. Interestingly, it has been mentioned that the main findings that suggest the development of AKI in patients with COVID-19 are acute tubular injury (ATI) or acute tubular necrosis (ATN) and collapsing glomerulopathy (CG) [12,27,28,29,30,31,32]. These findings were observed even before the visualization of viral particles in the kidney by electron microscopy or the detection of viral mRNA in the urine [12,27]. High-risk APOL1 alleles, a genetic risk factor for CG, could also affect disease susceptibility [29,33]. On the other hand, it has been suggested that patients with severe COVID-19, like patients with SARS and MERS, may have a cytokine storm syndrome with resulting exacerbated viral hyperinflammation characterized by increased IL-2, IL-7, G-CSF, CXCL10, MCP-1, MIP-1α, TNF-α, and IL-6 [34]. The cytokine storm cooperates with renal resident cells and promotes tubular and endothelial dysfunction. In this regard, activated lymphocytes migrate to renal tissues to destroy infected renal cells and release inflammatory cytokines, resulting in local inflammation and tissue damage [35]. Release of damage-associated molecular patterns (DAMPs) from injured tissue has been linked to organ-to-organ crosstalk and has also been suggested to mediate AKI in the context of acute respiratory distress syndrome (ARDS) [36]. Similarly, critically ill patients may be exposed to nephrotoxins that arise in response to all administered medications, such as antivirals, antibiotics, antifungals, and other nephrotoxic drugs, which can cause tubular injury or acute interstitial nephritis [37].

Available data suggest that the relationship between ACE2 and angiotensin II also contributes to kidney injury in COVID-19 [38]. Following SARS-CoV-2 binding to ACE2, ACE2 is thought to be downregulated, leading to increased angiotensin II levels and decreased Ang (1–7). These conditions favor the activation of the endothelium and platelets, vasoconstriction, and the release of proinflammatory cytokines [39]

Extracorporeal membrane oxygenation (ECMO) may also contribute to AKI by promoting venous congestion, increased risk of secondary infections, hemolysis, major bleeding, and inflammation [40]. Furthermore, excessive positive pressure ventilation administered to patients with COVID-19 potentially leads to adverse hemodynamic effects of decreased cardiac output, which amplifies renal hypoperfusion [41]. 

## 4. Clinical and Biochemical Manifestations of COVID-19 AKI

Several authors have described proteinuria and hematuria as two of the main initial signs in patients who develop COVID-19 AKI, although the proportion varies between studies. Tarragón B. et al. [42] reported proteinuria in 88.9% and hematuria in 79.4% of cases studied. Meanwhile, Chan L. et al. [43] describe that 84% of patients had proteinuria, 81% hematuria, and 60% leukocyturia. Blood urea nitrogen (BUN) and creatinine levels are elevated in patients with COVID-19 AKI, indicating kidney damage, and the hypercoagulable state caused by SARS-CoV-2 infection also results in thrombus and erythrocyte aggregation without obvious clots in the kidney [30]. Available reports indicate that rhabdomyolysis occurs in 7% to 20% of patients with evidence of AKI from COVID-19 [32,44]. Hyperkalemia has been observed in 23% of patients with COVID-19 AKI and is often associated with metabolic acidosis [44,45]. Although urinary kidney injury molecule 1 (uKIM-1) was shown to be a predictor of mortality in COVID-19 AKI [46] and damage at the level of the podocyte and renal tubule is important [24], multiple factors may be associated with damage at the same site, making it difficult at this time to separate the cytopathic component of the virus from other pre-existing and associated morbid conditions.

## 5. Risk Factors for the Development of COVID-19 AKI

The comorbidities most frequently associated with the development of COVID-19 AKI have been arterial hypertension (15–42.3%) [18,47], diabetes mellitus (7.4–41.4%) [31], and pre-existing chronic kidney disease (CKD) (0.7–7.6%) [18,47] (Table 1). Patients in the COVID-19 AKI group were significantly older than those in the non-AKI group, and AKI was found to be more prevalent among male patients (58.3%) [11]. Furthermore, COVID-19 AKI was higher in patients who had elevated serum creatinine levels at symptom onset than in patients with normal values [48]. In this same sense, it has been reported that COVID-19 AKI occurs later in patients who present a normal baseline serum creatinine at hospital admission, and they also recover more quickly [48].

Enikeev D. et al. [12] reported that severity of COVID-19 (OR = 23.09, CI: 7.89–67.57, *p* < 0.001) and history of CKD (OR = 7.17, CI: 2.09–24.47, *p* = 0.002) were strongly associated with the rate of COVID-19 AKI. Chan L. et al. [43] reported that independent predictors of severe COVID-19 AKI were CKD, male gender, and elevated serum potassium on admission. Chan L. et al. [43] reported that independent predictors of severe AKI were CKD, male gender, and elevated serum potassium at hospital admission. Meanwhile, Yildrim C. et al. [54] found that albuminuria and elevated serum cystatin C- and D-dimer levels at hospital admission could be early predictors of COVID-19 AKI. Other studies reported that elevated uKIM-1, which reflects proximal tubular injury, might be related to COVID-19 AKI and was positively correlated with risk of death [55,56]. Epidemiological data suggest that COVID-19 AKI is less common among patients in China [48,51,53] than among patients in the US [44,57,58] and Europe [13], although this difference can be attributed to the intrinsic characteristics of the population analyzed in the studies. For example, patients in the Chinese studies had fewer comorbidities and were hospitalized with less-severe respiratory illness than patients in other cohorts [13].

## 6. COVID-19 AKI Treatment

For the treatment of patients with COVID-19 AKI, a comprehensive treatment plan should be developed as soon as possible to avoid rapid deterioration of the patient’s condition. The implementation of the supportive care guidelines included in Kidney Disease: Improving Global Outcomes (KDIGO) is recommended in COVID-19 AKI patient management [59]. In this way, the treatment and management of patients with COVID-19 AKI is similar to that of patients with AKI associated with septic shock, being mainly supportive. Mitigation of volutrauma and barotrauma by applying lung-protective ventilation reduces the risk of developing or worsening AKI by limiting ventilation-induced hemodynamic effects on the kidney and bursts of cytokines [36]. To avoid renal congestion, another important option is to adjust fluid balance to avoid volume overload [60]. Although drug treatment can cause kidney damage, the amount of drugs administered can be adjusted appropriately to minimize this damage. The use of antiviral drugs such as ribavirin, oseltamivir, lopinavir/ritonavir, darunavir/cobicistat, arbidol, chloroquine, and hydroxychloroquine has been reported, but without clear evidence of renal benefit [61,62]. The same has happened with drugs that inhibit cytokine storms by various mechanisms, including immunosuppression by steroids or cyclosporine A [63], drugs directed against IL-6 such as tocilizumab and sturmuzumab [64,65], TNF-αs such as etanercept [65], IL-1βs such as anakinra [66], GCSFs such as lenzilumab [65], or IFNγs such as emapalumab [67]. Use of angiotensin-converting enzyme inhibitors and angiotensin-receptor blockers in patients with COVID-19 has been intensely debated but does not seem to affect outcomes [68].

If conservative management fails, renal replacement therapy (RRT) should be considered in patients with volume overload, especially those with refractory hypoxemia. Continuous RRT (CRRT) is the preferred modality in hemodynamically unstable patients with COVID-19 AKI. Overall, dialysis as CRRT was required in 0.8–64% of patients with COVID-19 AKI. CRRT blocks the production of inflammatory cascade reactions and downregulates body inflammation, removes toxins and waste products from damaged cells, and regulates hydro-electrolyte disturbance and acid–base balance, thus resulting in a stable and balanced internal environment [69]. It should be noted that the number or percentage of patients who recovered renal function after dialysis is variable. Different risk factors for death have been reported in patients with COVID-19 AKI receiving dialysis, including older age, severe illness, low lymphocyte count, high neutrophil count, pre-existing cardiovascular and/or renal disease, proteinuria, and hematuria [48]. Chan L. et al. [43] reported that 19% of patients with COVID-19 AKI required dialysis. The proportions with AKI stage 1, 2, or 3 were 39%, 19%, and 42%, respectively. Fisher M. et al. [10] reported a 49.5% prevalence of COVID-19 AKI in stage 1, 20.3% in stage 2, and 30.2% in stage 3, where 28.5% of patients with COVID-19 AKI stage 3 required RRT.

Notably, the decision to initiate acute RRT in patients with COVID-19 AKI should be individualized and should consider the clinical context (e.g., initiation of RRT for volume control in patients with severe hypoxemia) and not be based solely on the stage of AKI or the degree of kidney function [70]. Rizo-Topete LM. et al. [71] address dialysis indications in response to hemodynamic instability and fluid overload that are common in patients with COVID-19 AKI. Thus, they specify that CRRT should be the first treatment option if it is available, or peritoneal dialysis with an infusion volume of less than 25 mL/Kg. It is interesting that Sohaney R. et al. [72], in patients managed with continuous venovenous hemodiafiltration with regional citrate anticoagulation (CVVHDF-RCA), found no association between accumulated fluid balances and improvements in respiratory parameters.

Future studies are clearly required to analyze the relationship of therapies with clinicopathological characteristics and prognosis of COVID-19 AKI patients, but these are difficult since many factors must be considered. To mention a few, most patients having comorbidities are medicated before developing COVID-19, which can interfere with the results. Later, patients receive treatments according to their clinical characteristics and evolution, causing them to receive different doses and different drugs that may include antivirals, antihypertensives, antiglycemic agents, steroidal and non-steroidal anti-inflammatory drugs, anticoagulants, and occasionally experimental drugs that can interact with each other and cause variations in the observations. Indeed, drugs in the same family have different properties, potencies, and interactions. Third, variation in population, genetics, and environmental and nutritional characteristics of patients has important implications that must be considered.

## 7. COVID-19 AKI Mortality

Li Z. et al. [49] reported that patients with COVID-19 AKI had a ~5.3-fold higher mortality risk than those without AKI. This is higher than other comorbid chronic diseases. Caceres PS. et al. [19] reported that higher viral load was correlated with mortality, but not with albuminuria or AKI stage. Furthermore, Xiao G. et al. [50] showed that mortality in AKI stage 1 was 7.3% while in stage 2 and 3 it was 64.3% at 28 days. Along with this, Tarragon B. et al. [42] reported that the absolute risk of in-hospital mortality was significantly increased (*p* values < 0.0001) with each raise of the AKI stage compared to the control group without AKI. Overall, the OR for hospital mortality in those with AKI compared to those without AKI was 4 (95% CI 3.5 to 4.5).

Roushani J. et al. [73] reported a mortality of 58% in patients with COVID-19 ARI at 30 days after starting RRT and 64% at 90 days. At 90 days, 20% of survivors were still dependent on RRT and 31% were still hospitalized. Chan L. et al. [43] reported a 46% prevalence of AKI due to COVID-19 and a 50% hospital mortality in these. Of discharged AKI survivors, 35% had not recovered baseline renal function at discharge. Moledina DG. et al. [74] reported that 8.5% of patients with COVID-19 AKI required dialysis and had lower recovery rates. Meanwhile, Ng JH. et al. [75] reported that the risk of in-hospital death was increased in COVID-19 AKI with a relative risk for AKI stage 1–3 of approximately 3.4 (95% CI: 3.0–3.9) and for AKI stage 3 receiving dialysis of 6.4 (95% CI: 5.5–7.6) compared to those without AKI. Interestingly, among survivors with AKI stage 1–3, 74.1% achieved renal recovery at discharge, while among survivors with AKI stage 3 receiving dialysis, 30.6% remained on dialysis at discharge. Prehospital chronic kidney disease was the only independent risk factor associated with the need for dialysis at discharge. Other authors report an in-hospital death of 33.7% in those with COVID-19 AKI compared to 9.3% in those without AKI [10].

Recently, Samaan F. et al. [76] reported in a retrospective cohort of 375 COVID-19 AKI patients receiving RRT a mortality rate of 72.5%, and a large percentage of surviving patients (22.3%) continued RRT for an extended period of time. Similarly, Gupta S. et al. [77] reported that 20.6% of patients with COVID-19 AKI required RRT within 14 days of ICU admission, and 54.9% of these patients died within 28 days after admission. In addition, 18.1% of patients with COVID-19 AKI who were discharged still required RRT at 60 days.

## 8. Conclusions

AKI is common among patients with SARS-CoV-2 infection, and it is frequently associated with the need for replacement therapy and maintenance dialysis, which reduces the quality of life of patients. Notably, COVID-19 AKI is also associated with a high mortality rate. For all these reasons, more focused research is required in this area in order to find clearer answers to how SARS-CoV-2 infects the kidney and renal cells, the pathognomonic signs of infection and the mechanisms of damage, and how to prevent or avoid it. Whether different variants of the virus have different effects on the kidney or whether vaccines prevent kidney damage has not yet been investigated. It would also be advisable to find suitable and, if possible, low-cost clinical tests for timely diagnosis before kidney function is compromised.

## Figures and Tables

**Figure 1 diseases-10-00053-f001:**
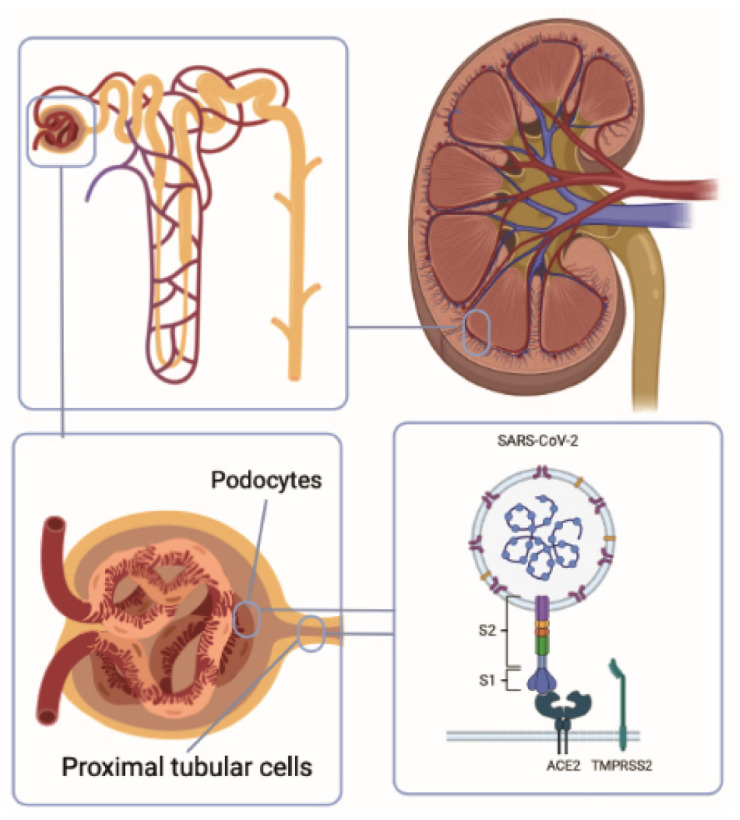
SARS-CoV-2 infects the kidney. The kidney is one of the organs that most expresses ACE2, the receptor that SARS-CoV-2 uses to enter cells. The renal cells with the highest expression are proximal tubular cells and podocytes.

**Figure 2 diseases-10-00053-f002:**
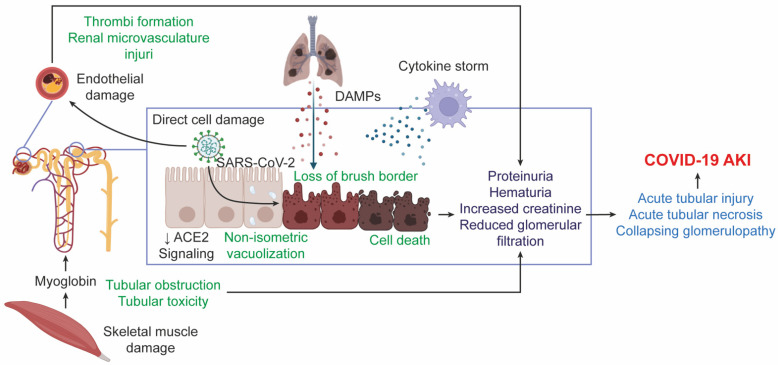
Hypothesis of the COVID-19 AKI pathogenesis. SARS-CoV-2 can cause kidney damage directly or indirectly. Cytokine storm and DAMP secretion from damaged tissues, such as the lung, may contribute to cell death of renal cells such as podocytes and proximal tubular cells. In addition, the prothrombotic state can facilitate the generation of clots in the nephron, which contribute to damage. All this together causes patients who develop COVID-19 AKI to present with proteinuria, hematuria, increased creatinine, and decreased glomerular filtration rate. Acute tubular injury or acute tubular necrosis and collapsing glomerulopathy are generated, which can evolve into COVID-19 AKI.

**Table 1 diseases-10-00053-t001:** Main characteristics observed in COVID-19 AKI reports.

Reference	n	Males	Average Age (Years)	AKI (Stage)	Need CRRT	Probability of Death (HR and CI)	Antibiotic Therapy (All Patients/AKI Patients)	Antiviral Drugs (All Patients/AKI Patients)	Antifungal Drugs (All Patients/AKI Patients)	Glucocorticoids (All Patients/AKI Patients)	Underlying Diseases
[9]	116	67	54	None	4.3% (previous CKD)	None	NM	NM	NM	NM	Hypertension 37.1% Diabetes 15.5% Malignant tumor 10.3% CKD on hemodialysis 4.3% Cerebral infarction 6%
[49]	193	95	57	28.4%	4%	NM	NM	98%/NM	NM	62%/NM	Cerebrovascular disease 36% Endocrine system disease 20% Cardiovascular and respiratory system disease 10% Urinary system disease 5% Nervous system disease 3% Reproductive system disease 2%
[10]	4610, 3345 COVID-19 positive patients, 1265 COVID-19 negative patients	Total positive for COVID-19 sex male 53.1%	Total positive for COVID-19 64.4	AKI in 1903 COVID-19 positive patients; AKI 1: 49.5%, AKI 2: 20.3%, AKI 3: 30.2%	28.5% of patients with stage 3 AKI	Death of patients COVID-19 positive compared to those negative for COVID-19 (23.2% versus 7.3%; RR, 3.8; 95% CI, 2.6 to 3.9)	NM	NM	NM	NM	Total positive for COVID-19: Diabetes 27.1%, CKD 12.2%, lung disease 4.9%, malignancy 1.8%
[48]	701	367	63	5.1% AKI 1 (1.9%) AKI 2 (1.3%) AKI 3 (2%)	NM	All: 2.1%, 95% CI (1.36–3.26) AKI 1 (1.9%, 0.76–0.476) AKI 2 (3.51%, 1.49–8.26) AKI 3 (4.38%, 2.31–8.31)	71%/75%	73%/58.3% *	NM	36.9%/58.3% *	Hypertension 33.4% Diabetes 14.3% Tumor 4.6% CKD 2% COPD 1.9%
[50]	287	160	62	19.2% AKI 1 14.3%, AKI 2–3 4.9%.	NM	NM	NM	NM	NM	NM	Hypertension 30% Diabetes mellitus 16% Cardiovascular disease 12% Cerebrovascular disease 8% COPD 6% Chronic liver disease 4% CKD 2% Cancer 3%
[48]	710	374	63	3.2% AKI 1 (1.1%) AKI 2 (0.8%) y AKI 3 (1.1%)	NM	2.21 (95% CI: 1.11–4.39)	NM	NM	NM	NM	NM
[18]	26 autopsies	19	69	NM	19.2%	NM	NM	61.5%/NM	NM	61.5%/NM	Hypertension 42.3% History any cancer 23% Diabetes mellitus 11.5% CKD 7.6 %
[47]	41	30	49	7%	7%	NM	100%/NM	93%/NM	NM	22%/NM	Diabetes 20%, Hypertension 15%, Cardiovascular disease 15%, COPD 2%, Malignancy 2%, Chronic liver disease 2%
[51]	138	75	56	3.6%	1.45%	NM	NM/NM	89.9%/NM	NM/NM	44.9%/NM	Hypertension 31.2% Cardiovascular disease 14.5% Diabetes 10.1% Malignancy 7.2%, Cerebrovascular disease 5.1% COPD 2.9% CKD 2.9% Chronic liver disease 2.9% HIV infection 1.4%
[31]	1099	640	47	0.5%	0.8%	NM	58%/NM	41.3%/NM	2.8%/NM	18.6%/NM	Hypertension 15% Diabetes 7.4% Coronary heart disease 2.5% Hepatitis B infection 2.1% Cerebrovascular disease 1.4% COPD 1.1% Any type of cancer 0.9% CKD 0.7% Immunodeficiency 0.2%
[52]	274	171	62	10.5%	NM	NM	91%/NM	86%/NM	NM/NM	79%/NM	Hypertension 34.3% Diabetes 17.1% Cardiovascular disease 8.3% Malignancy 2.5% CKD 1.4% Cerebrovascular disease 1.4% Gastrointestinal disease 1%
[53]	99	67	55.5	3%	9%	NM	71%/NM	76%/NM	15%/NM	19%/NM	Cardiovascular and cerebrovascular disease 40% Digestive system disease 11% Endocrine system disease 11% Malignant tumor 1% Nervous system disease 1% Respiratory system disease 1%

AKI: acute kidney injury, HR: hazard ratio, CI: confidence interval, CRRT: continuous renal replacement therapy, CKD: chronic kidney disease, COPD: chronic obstructive pulmonary disease, NM: Not mentioned. * *p* value < 0.05 with AKI and no-AKI patients.

## Data Availability

Data were not used nor created for this review. Figures were created using Biorender.com (accessed on 18 August 2022).

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
