# Peer review of "Acute Kidney Injury in COVID-19 Patients: Pathogenesis, Clinical Characteristics, Therapy, and Mortality"

_diseases, 2022, doi:10.3390/diseases10030053_

Round 1

Reviewer 1 Report

There is a considerable number of patients with COVID-19 virus infection were suffered from AKI. To review the pathogenesis, clinical characteristics, therapy and mortality of COVID-19 could help us further understand the kidney injury induced by COVID-19 virus and associated therapies including antibiotics, antiviral drugs, anti-fungal drugs. But the paper written by Venice and other authors simply described the general information, thus could not be published before a major revision.

I suggest that the important revision should include: 

1.Further collect the information about the basic studies of the causes of AKI and relative mechanisms induced by COVID-19 virus infection as well as the interventions of COVID-19.

2. Introduce the therapies except RRT and discuss the relationship between clinical pathological characteristics, interventions and prognosis of AKI in patients with COVID-19.

Author Response

Dear Reviewer,

We appreciate your help in improving our work, responses to your comments appear below in blue letters and highlighted in red letters in the manuscript.

Reviewer 1

There is a considerable number of patients with COVID-19 virus infection were suffered from AKI. To review the pathogenesis, clinical characteristics, therapy and mortality of COVID-19 could help us further understand the kidney injury induced by COVID-19 virus and associated therapies including antibiotics, antiviral drugs, anti-fungal drugs. But the paper written by Venice and other authors simply described the general information, thus could not be published before a major revision.

I suggest that the important revision should include: 

1.Further collect the information about the basic studies of the causes of AKI and relative mechanisms induced by COVID-19 virus infection as well as the interventions of COVID-19.

We added a section titled “3. Pathophysiology of COVID-19 AKI”. Throughout the manuscript we add more information on the mechanisms used by SARS-CoV-2 to cause the development of COVID-19 AKI. In the manuscript it appears as follows:

  1. Pathophysiology of COVID-19 AKI

The pathophysiology of COVID-19 AKI is thought to be multifactorial and involves cardi-ovascular comorbidity, direct effects of the virus on the kidney, local and systemic inflammatory and immune responses, endothelial injury, and activation of coagulation pathways and the ren-in-angiotensin system (Figure 2) [23]. Pan X. et al (2020) [24] mentioned that the cytopathic effects of SARS-CoV-2 on podocytes and proximal tubule cells might be associated with the development of COVID-19 AKI, especially in patients with sepsis. Likewise, the presence of viral particles in renal endothelial cells has been reported, suggesting that viremia may also cause endothelial damage that promotes vasoconstriction, a state of hypercoagulability, and macrophage activa-tion leading to microthrombi formation and renal microvasculature injury [18,24]. Furthermore, the loss of functional nephrons after injury could increase the development of renal fibrosis. Meanwhile, COVID-19 pneumonia can cause right ventricular failure and lead to renal conges-tion, while left ventricular dysfunction can lead to hypotension, decreased cardiac output, and hypoperfusion of the kidneys [25]. Severe COVID-19 can cause skeletal muscle damage leading to myoglobulin release, which induces kidney damage through pigment cast formation causing tub-ular obstruction and tubular toxicity related to iron release [26].

In clinical studies, COVID-19 AKI has been associated first with sepsis (cytokine storm) and septic shock (hypoxia), second with the use of nephrotoxic drugs, and third with direct cell damage caused by SARS-CoV-2 [16,17]. Histopathological reports of AKI and COVID-19 are scarce and are usually autopsy series or isolated reports of some glomerulopathy. However, Su H. et al (2020) [18] reported pathologic renal abnormalities in post-mortem cases, demonstrating the presence of coronavirus clusters in the proximal tubular epithelium, podocytes, and distal tubules along with loss of brush border and non-isometric vacuolization. Interestingly, it has been mentioned that the main findings that suggest the development of AKI in patients with COVID-19 are acute tubular injury (ATI) or acute tubular necrosis (ATN) and collapsing glomerulopathy (CG) [12,27–32]. These findings were observed even before the visualization of viral particles in the kidney by electron microscopy or the detection of viral mRNA in the urine [12,27]. High-risk APOL1 alleles, a genetic risk factor for CG, could also affect disease suscepti-bility [29,33]. On the other hand, it has been suggested that patients with severe COVID-19, like patients with SARS and MERS, may have a cytokine storm syndrome with resulting exacerbated viral hyperinflammation characterized by increased IL-2, IL-7, G-CSF , CXCL10, MCP-1 , MIP-1α, TNF-α and IL-6 [34]. The cytokine storm cooperates with renal resident cells and promotes tubular and endothelial dysfunction. In this regard, activated lymphocytes migrate to renal tissues to destroy infected renal cells and release inflammatory cytokines, resulting in local inflammation and tissue damage [35]. Release of damage-associated molecular patterns (DAMPs) from injured tissue has been linked to organ-to-organ crosstalk and has also been suggested to mediate AKI in the context of acute respiratory distress syndrome (ARDS) [36]. Similarly, critically ill patients may be exposed to nephrotoxins that arise in response to all administered medications, such as antibiotics, which can cause tubular injury or acute interstitial nephritis [37].

Available data suggest that the relationship between ACE2 and angiotensin II also contributes to kidney injury in COVID-19 [38]. Following SARS-CoV-2 binding to ACE2, ACE2 is thought to be downregulated, leading to increased angiotensin II levels and decreased Ang(1–7). These conditions favor the activation of the endothelium and platelets, vasoconstriction and the release of proinflammatory cytokines [39]

Extracorporeal membrane oxygenation (ECMO) may also contribute to AKI by promoting venous congestion, increased risk of secondary infections, hemolysis, major bleeding, and inflammation [40]. Furthermore, excessive positive pressure ventilation administered to patients with COVID-19 potentially leads to adverse hemodynamic effects of decreased cardiac output, which amplifies renal hypoperfusion [41].

  1. Introduce the therapies except RRT and discuss the relationship between clinical pathological characteristics, interventions and prognosis of AKI in patients with COVID-19.

The title of section 5 was changed to “5. COVID-19 AKI treatment” and other therapies used in the treatment of COVID-19 AKI, in addition to RTT, and their relationship with clinicopathological characteristics and prognosis were discussed. In the manuscript it appears as follows:

  1. COVID-19 AKI treatment

For the treatment of patients with COVID-19 AKI, a comprehensive treatment plan should be developed as soon as possible to avoid rapid deterioration of the patient's condition. The implementation of the supportive care guidelines included in the Kidney Disease: Improving Global Outcomes (KDIGO) is recommended in COVID-19 AKI patient management [58]. In this way, the treatment and management of patients with COVID-19 AKI is similar to that of patients with AKI associated with septic shock, being mainly supportive. Mitigation of volutrauma and barotrauma by applying lung-protective ventilation reduces the risk of developing or worsening AKI by limiting ventilation-induced hemodynamic effects on the kidney and burst of cytokines [36]. To avoid renal congestion, another important option is to adjust fluid balance to avoid volume overload [59]. Although drug treatment can cause kidney damage, the amount of drugs administered can be adjusted appropriately to minimize this damage. The use of antiviral drugs such as ribavirin, oseltamivir, lopinavir/ritonavir, darunavir/cobicistat, arbidol, chloroquine and hydroxychloroquine has been reported, but without clear evidence of renal benefit [60,61]. The same has happened with drugs that inhibit cytokine storms by various mechanisms, including immunosuppression by steroids or cyclosporine A [62], drugs directed against IL-6 such as tocilizumab and sturmuzumab [63,64], TNF-α such as etanercept [64], IL-1β such as anakinra [65], GCSF such as lenzilumab [64], or IFNγ such as emapalumab [66]. Angiotensin-converting enzyme inhibitors and angiotensin-receptor blockers, in patients with COVID-19 has been intensely debated but does not seem to affect outcomes [67].

If conservative management fails, renal replacement therapy (RRT) should be considered in patients with volume overload, especially those with refractory hypoxemia. Continuous RRT (CRRT) is the preferred modality in haemodynamically unstable patients with COVID-19 AKI. Overall, dialysis as CRRT was required in 0.8-64% of patients with COVID-19 AKI. CRRT blocks the production of inflammatory cascade reactions and downregulates body inflammation, removes toxins and waste products from damaged cells, regulates hydroelectrolyte disturbance and acid-base balance, thus resulting in a stable and balanced internal environment [68]. It should be noted that the number or percentage of patients who recovered renal function after dialysis is variable. Different risk factors for death have been reported in patients with COVID-19 AKI receiving dialysis, including older age, severe illness, low lymphocyte count, high neutrophil count, pre-existing cardiovascular and/or renal disease, proteinuria, and hematuria [48]. Chan L. et al. [43] reported that 19% of patients with COVID-19 AKI required dialysis. The proportions with AKI stage 1, 2, or 3 were 39%, 19%, and 42%, respectively. Fisher M. et al (2020) [10] reported a 49.5% prevalence of COVID-19 AKI in stage 1, 20.3% in stage 2, 30.2% in stage 3, where 28.5% of patients with COVID-19 AKI stage 3 required RRT.

Notably, the decision to initiate acute RRT in patients with COVID-19 AKI should be individualized and should consider the clinical context (e. g. initiation of RRT for volume control in patients with severe hypoxemia) and not be based solely on in the stage of AKI or the degree of kidney function [69]. Rizo-Topete LM. et al (2020) [70] address dialysis indications in response to hemodynamic instability and fluid overload that are common in patients with COVID-19 AKI. Thus, they specify that CRRT should be the first treatment option if it is available or peritoneal dialysis with an infusion volume of less than 25 mL/Kg. It is interesting that Sohaney R. et al (2021) [71] in patients managed with continuous venovenous haemodiafiltration with regional citrate anticoagulation (CVVHDF-RCA) found no association between accumulated fluid balances and improvements in respiratory parameters.

Future studies are clearly required to analyze the relationship of therapies with clinicopathological characteristics and prognosis of COVID-19 AKI patients, but these are difficult since many factors must be considered. To mention a few, most patients, having comorbidities, are medicated before developing COVID-19, which can interfere with the results. Later, patients receive treatments according to their clinical characteristics and evolution, causing them to receive different doses and different drugs that may include antivirals, antihypertensives, antiglycemic agents, steroidal and non-steroidal anti-inflammatory drugs, anticoagulants, and occasionally experimental drugs that can interact with each other and cause variations in the observations. Indeed, drugs in the same family have different properties, potencies, and interactions. Third, variation in population, genetic, environmental, and nutritional characteristics of patients has important implications that must be considered.

Reviewer 2 Report

The publication is an important part of understanding various aspects of complications following SARS-Cov-2 infection, which often occur after the completion of basic treatment. Acute renal failure (AKI) is a very dangerous complication for patients' life and health, which not only affects the quality of life but also significantly increases the costs of health care in many countries. Knowledge about this disease's aetiology and course may contribute to improving care for patients after COViD-19. This paper presents the current state of knowledge about this complication clearly and lucidly.

Author Response

Dear Reviewer,

We appreciate your help in improving our work, responses to your comments appear below in blue letters and highlighted in red letters in the manuscript.

Reviewer 2

The publication is an important part of understanding various aspects of complications following SARS-Cov-2 infection, which often occur after the completion of basic treatment. Acute renal failure (AKI) is a very dangerous complication for patients' life and health, which not only affects the quality of life but also significantly increases the costs of health care in many countries. Knowledge about this disease's aetiology and course may contribute to improving care for patients after COViD-19. This paper presents the current state of knowledge about this complication clearly and lucidly.

We appreciate your comments

Reviewer 3 Report

In this article authors reviewed the issue of AKI among patients with COVID 19 infections. The article is interesting and well explained.

My only comment is whether the authors think it is possible to explain from the data highlighted, how the component of renal failure linked to viral damage can be differentiated from that linked to other factors (pharmacological damage, dehydration etc)

Author Response

Dear Reviewer,

We appreciate your help in improving our work, responses to your comments appear below in blue letters and highlighted in red letters in the manuscript.

Reviewer 3

In this article authors reviewed the issue of AKI among patients with COVID 19 infections. The article is interesting and well explained

My only comment is whether the authors think it is possible to explain from the data highlighted, how the component of renal failure linked to viral damage can be differentiated from that linked to other factors (pharmacological damage, dehydration etc).

Although U-KIM1 was shown to be a predictor of mortality in COVID-19 AKI and damage at the level of the podocyte and renal tubule is important, multiple factors may be associated with damage at the same site, making it difficult at this time to separate the cytopathic component of the virus from other pre-existing and associated morbid conditions. This is mentioned in the manuscript.

Round 2

Reviewer 1 Report

The adding information could describe clearly the related contents about the AKI in patients with COVID-19 infection. I suggest the paper could be published.